# Serotonin Receptor 5-HT_2A_ Regulates TrkB Receptor Function in Heteroreceptor Complexes

**DOI:** 10.3390/cells11152384

**Published:** 2022-08-02

**Authors:** Tatiana Ilchibaeva, Anton Tsybko, Andre Zeug, Franziska E. Müller, Daria Guseva, Stephan Bischoff, Evgeni Ponimaskin, Vladimir Naumenko

**Affiliations:** 1Cellular Neurophysiology, Center of Physiology, Hannover Medical School, Carl-Neuberg Strasse 1, 30625 Hannover, Germany; zeug.andre@mh-hannover.de (A.Z.); mueller.franziska@mh-hannover.de (F.E.M.); 2Laboratory of Behavioral Neurogenomics, Federal Research Center Institute of Cytology and Genetics, Siberian Branch of Russian Academy of Sciences, Prospekt Lavrentyeva 10, 630090 Novosibirsk, Russia; antontsybko@bionet.nsc.ru (A.T.); naumenko2002@mail.ru (V.N.); 3Department of Nutritional Medicine, University of Hohenheim, Fruwirthstr. 12, 70599 Stuttgart, Germany; daria.guseva@uni-hohenheim.de (D.G.); bischoff.stephan@uni-hohenheim.de (S.B.)

**Keywords:** tropomyosin receptor kinase B, 5-hydroxytryptamine 2A receptor, oligomerization, heteroreceptor, autophosphorylation

## Abstract

Serotonin receptor 5-HT_2A_ and tropomyosin receptor kinase B (TrkB) strongly contribute to neuroplasticity regulation and are implicated in numerous neuronal disorders. Here, we demonstrate a physical interaction between 5-HT_2A_ and TrkB in vitro and in vivo using co-immunoprecipitation and biophysical and biochemical approaches. Heterodimerization decreased TrkB autophosphorylation, preventing its activation with agonist 7,8-DHF, even with low 5-HT_2A_ receptor expression. A blockade of 5-HT_2A_ receptor with the preferential antagonist ketanserin prevented the receptor-mediated downregulation of TrkB phosphorylation without restoring the TrkB response to its agonist 7,8-DHF in vitro. In adult mice, intraperitoneal ketanserin injection increased basal TrkB phosphorylation in the frontal cortex and hippocampus, which is in accordance with our findings demonstrating the prevalence of 5-HT_2A_–TrkB heteroreceptor complexes in these brain regions. An expression analysis revealed strong developmental regulation of 5-HT_2A_ and TrkB expressions in the cortex, hippocampus, and especially the striatum, demonstrating that the balance between TrkB and 5-HT_2A_ may shift in certain brain regions during postnatal development. Our data reveal the functional role of 5-HT_2A_–TrkB receptor heterodimerization and suggest that the regulated expression of 5-HT_2A_ and TrkB is a molecular mechanism for the brain-region-specific modulation of TrkB functions during development and under pathophysiological conditions.

## 1. Introduction

Tropomyosin receptor kinase B (TrkB) is a major transducer of intracellular signals from mature brain-derived neurotrophic factor (BDNF). TrkB is predominantly expressed within the CNS, especially in the frontal cortex, hippocampus, cerebellar cortex, visual system, hypothalamus, striatum, substantia nigra, and dorsal raphe nucleus [1,2,3,4,5,6]. The binding of BDNF to TrkB induces the autophosphorylation of tyrosine residues in the cytoplasmic domain of TrkB, resulting in the activation of the Ras–mitogen-activated protein kinase (MAPK), phosphoinositide 3-kinase (PI3K)–AKT, and phospholipase Cγ1 (PLC-γ1) signaling pathways [7,8]. BDNF–TrkB signaling plays important roles in the regulation of neuron survival and migration, neurite growth, and long-term potentiation [8,9,10]. Additionally, TrkB-mediated signaling is broadly recognized as a critical component of the responses to classic and unconventional antidepressants [11,12].

Serotonin receptor 2A (5-HT_2A_) belongs to the G-protein-coupled receptor (GPCR) family and is broadly distributed within the brain, showing high expression in the olfactory tubercle, cortical areas, and dentate gyrus [13,14]. Functionally, 5-HT_2A_ is involved in neurogenesis and can modulate synaptic plasticity via long-lasting increases in the excitability and firing rate of glutamatergic and GABAergic neurons [15,16]. The 5-HT_2A_ receptor is implicated in numerous CNS disorders, including bipolar disorder, depression, and schizophrenia [17,18]. A number of open-label and placebo-controlled clinical trials have revealed that some 5-HT_2A_-blocking antipsychotic drugs elicit better clinical responses in bipolar or treatment-resistant depressive patients [19,20,21,22,23,24].

The coupling of 5-HT_2A_ with heterotrimeric G_q_/G_11_ proteins results in the activation of phospholipase C (PLC), leading to increased Ca^2+^ release from the endoplasmic reticulum. In neurons, Ca^2+^ influx can activate cyclic AMP (cAMP) response element-binding protein (CREB), which, in turn, boosts the transcription of BDNF [25]. Interestingly, a number of studies show increased BDNF expression following 5-HT_2A_ stimulation [26,27,28,29], suggesting functional interplay between the 5-HT_2A_ and TrkB signaling pathways. Although the underlying mechanisms are not completely understood, a direct interaction between 5-HT_2A_ and TrkB may be an intriguing explanation. Research groups have demonstrated that 5-HT_2A_ can form heterodimers with other GPCRs, including the 5-HT_1A_ receptor [30], dopamine receptor D2 [31], and metabotropic glutamate receptor mGlu(2) [32]. Moreover, we have previously reported that other 5-HT receptors, including the 5-HT_4_ and 5-HT_7_ subtypes, can form heterodimers with adhesion molecule L1 [33], CDK5 [34], or CD44 [35], respectively. To date, no published data show that 5-HT_2A_ interacts with non-GPCR proteins.

In the present study, we found that 5-HT_2A_ and TrkB form heteroreceptor complexes both in vitro and in vivo. We further demonstrated the functional effects of 5-HT_2A_ and TrkB receptor heterodimerization.

## 2. Materials and Methods

### 2.1. Animals and Drugs

Adult (P60) male C57BL/6J mice (10–11 weeks old, 23–26 g, Jackson Laboratory, Bar Harbor, ME, USA, RRID:IMSR_JAX:000664) were used for acute treatment with ketanserin or 25CN-NBOH. The mice were housed under standard laboratory conditions in the natural light–dark cycle (16 h of light and 8 h of dark) with free access to water and food. The preferential 5-HT_2A_ antagonist ketanserin (Sigma Aldrich, St. Louis, MO, USA) or the selective 5-HT_2A_ agonist 25CN-NBOH (Tocris, Bristol, UK) was dissolved in saline, and each was administered i.p. at a dose of 1 mg/kg of body weight (eight mice in each group). The control group (eight mice) received an equivalent saline injection. At 10 min (for 25CN-NBOH) or 30 min (for ketanserin) after the injection, the animals were euthanized, the brains were removed on ice, and the frontal cortex, hippocampus, and striatum were dissected.

### 2.2. Cell Culture and Transfection

Mouse N1E-115 neuroblastoma cells from the American Type Culture Collection (ATCC, cat. # CRL-2263, RRID:CVCL_0451) were grown at 37 °C and 5% CO_2_ in DMEM (Gibco, New York, NY, USA) that contained 10% of fetal bovine serum (Gibco, New York, NY, USA) and penicillin/streptomycin (100 U/mL, Gibco). Transient transfection was performed with Lipofectamine 2000 (Invitrogen) according to manufacturer’s protocol, and after transfection cells were incubated in the serum-free medium. For treatments, we applied 500 nM selective TrkB agonist 7,8-DHF (Sigma Aldrich, St. Louis, MO, USA), 1 μM 25CN-NBOH (Tocris, Bristol, UK), and/or 1 μM ketanserin (Sigma Aldrich, St. Louis, MO, USA). The doses of all drugs were chosen based on literature data. In one work, it was shown that a 7,8-DHF concentration of 500 nM is maximally effective for TrkB activation [36]. 25CN-NBOH at 1 μM is highly selective for 5-HT_2_ sites [37]. Ketanserin at 1 μM is known to block the 5-HT_2A_ receptor [38].

### 2.3. Recombinant-DNA Procedures

Murine 5-HT_2A_ cDNA was cloned into the pcDNA3.1(+) donor vector (Invitrogen, Carlsbad, CA, USA) carrying an HA-tag. TrkB_mEGFP was kindly gifted by Ryohei Yasuda (Addgene plasmid # 83952; http://n2t.net/addgene:83952 (accessed on 13 July 2019); RRID: Addgene_83952).

### 2.4. Co-Immunoprecipitation

Co-immunoprecipitation in lysates of N1E-115 cells that co-expressed HA-tagged 5-HT_2A_ and GFP-tagged TrkB was performed as described previously [35] with an antibody against GFP (1:250; GeneTex, cat. # GTX26556, RRID:AB_371421). Immunoblotting was carried out using a horseradish peroxidase (HRP)-conjugated anti-GFP (1:1000; LSBio (LifeSpan, Providence, RI, USA), cat. # LS-C50850-500, RRID:AB_1220053) or an HRP-conjugated anti-HA tag (1:250; Roche, Basel, Switzerland, cat. # 12013819001, RRID:AB_390917) antibody.

Co-immunoprecipitation from hippocampal, cortical, and striatal homogenates was performed according to a protocol described elsewhere [39]. Briefly, brain samples isolated from adult (P90) C57BL/6J mice were homogenized, and membrane fractions were prepared by differential centrifugation. The lysates were incubated with a goat polyclonal antibody against 5-HT_2A_ (1:50; Santa Cruz Biotechnology, Dallas, TX, USA, cat. # sc-15073, RRID:AB_2119724) followed by incubation with Protein A-Sepharose and Western blotting with an anti-TrkB antibody (1:1000; R&D Systems, Minneapolis, MN, USA, cat. # AF1494, RRID:AB_2155264).

### 2.5. The Immunofluorescence Assay of Mouse Brain Sections

Adult (P90) male and female C57BL/6J mice were subjected to this assay. The mice were bred and housed at the animal facility of the University of Hohenheim at a controlled temperature (25 °C) and photoperiod (12/12 h light/dark cycle) and were allowed unrestricted access to standard food and tap water.

Mice were euthanized by CO_2_ asphyxiation, the brains were removed, postfixed overnight at 4 °C, processed routinely, and embedded in paraffin. Next, 5 µm-thick sections were prepared on a Leica RM2255 microtome (Nussloch, Germany) and transferred to SuperFrost^®^Plus glass slides (Thermo Fisher Scientific, Waltham, MA, USA). The sections were air-dried at least overnight at 37 °C and then subjected to the immunofluorescence assay.

The brain sections were deparaffinized according to a standard protocol and washed with PBS. Antigen retrieval was performed by immersion in 0.01 M sodium citrate buffer (pH 9.0) heated at 80 °C in a water bath for 30 min. The blocking of nonspecific binding sites was performed for 1 h at room temperature in PBS containing 5% normal donkey serum (Jackson ImmunoResearch Laboratories, West Grove, PA, USA, cat. # 017-000-121), 0.1% Triton X-100 (Sigma-Aldrich, Darmstadt, Germany, cat. # 93427), and 0.02% sodium azide (Merck, Darmstadt, Germany). Incubation with rabbit polyclonal anti-5-HT_2A_ receptor (1:100, Abcam, Cambridge, UK, cat. # ab66049, RRID:AB_1141522) and goat polyclonal anti-TrkB (1:50, R&D Systems, cat. # AF1494, RRID:AB_2155264) antibodies diluted in PBS was carried out overnight at 4 °C. After a wash in PBS, appropriate secondary antibodies were applied for a 1 h incubation at room temperature: a donkey anti-rabbit IgG antibody conjugated with Alexa Fluor^®^ 488 (Jackson ImmunoResearch Labs, RRID:AB_2313584, cat. # 711-545-152) and a donkey anti-goat IgG antibody conjugated with Alexa Fluor^®^ 594 (Jackson ImmunoResearch Labs, RRID:AB_2340433, cat. # 705-585-147) both diluted 1:800 in PBS. After a subsequent wash in PBS, cell nuclei were visualized with a bis-benzimide solution (Hoechst 33,258 dye, 5 µg/mL in PBS; Sigma-Aldrich, St. Louis, MO, USA). Finally, the sections were mounted in an antiquenching medium and examined under a fluorescence microscope Zeiss Axiovert 200M (Zeiss, Göttingen, Germany) with 20× air objectives.

### 2.6. Linear Unmixing FRET Analysis

These measurements were performed on live N1E-115 neuroblastoma cells as described elsewhere [40]. 5-HT_2A_-mTurquoise2 and TrkB-YPet served as the donor and acceptor, respectively. As a negative control, cells were co-transfected with mTurquoise2-tagged CD86 and YPet-tagged CD86. After 16 h, the cells were imaged with a Zeiss LSM 780 microscope equipped with a C-Apochromat 40×/1.2 W Korr water immersion objective via the excitation of the fluorescent proteins at 440 and 514 nm according to the protocol. For the image analysis and evaluation, custom-written MATLAB scripts were employed. We calculated the predicted apparent FRET efficiency as Ef_DA_ = ½ Ef_D_/(1 − x_D_) = ½ Ef_A_/x_D_, assuming a standard dimerization model [39,41].

### 2.7. Analysis of Ca^2+^ Activity

This assay was performed on N1E-115 cells expressing TrkB-YPet, 5-HT_2A_-mTurquoise2, or both. We acquired time series data under the Zeiss LSM 780 microscope for 10 min per recording (5 s per frame). Ca^2+^ activity was assessed by means of GCaMP6f fluorescence signals (F) and changes calculated as F/F_max_. To raise Ca^2+^ levels to saturated, 10 µM ionomycin was applied after 6 min. This saturated Ca^2+^ signal was used as F_max_ for the calculation of basal Ca^2+^ levels.

### 2.8. In Situ Proximity Ligation Assay (PLA)

A PLA assay was performed with the Duolink in situ PLA Probes and Duolink in situ Detection Reagents Red Kit (cat. # DUO92008). Brain sections of 20 μm thickness were utilized in the PLA. Antibodies, the same as those used for immunohistochemistry, were diluted 1:100 with the Antibody Diluent provided with the kit. For imaging, the slides were dried and mounted with a cover slip by means of ~7 μL of the Duolink in situ Mounting Medium with 4′,6-diamidino-2-phenylindole (DAPI). The imaging and analysis of the stained brain sections were carried out under the fluorescence confocal microscope (Zeiss LSM 780) using a 40× objective and excitation wavelengths of 561 nm for PLA and 405 nm for DAPI. In each brain section, 3–4 measurements in the frontal cortex, hippocampus, and striatum were performed. Each image was analyzed in ImageJ (Fiji, RRID:SCR_002285) to count individual fluorescent spots. The data are presented as the number of spots per cell, normalized to the number of cells.

### 2.9. qRT-PCR

Total RNA was extracted from the brain tissue of C57BL/6J mice using ExtractRNA (Evrogen, Moscow, Russia), treated with RNA-free DNase (Promega, Madison, WI, USA), and diluted to 0.125 µg/µL with diethyl pyrocarbonate-treated water. One microgram of total RNA was subjected to cDNA synthesis with a random hexanucleotide mixture [42,43,44]. The number of cDNA copies for all studied genes was evaluated by qPCR on a LightCycler 480 (Roche Applied Science, Rotkreuz, Switzerland) with specific primers (Table 1), SYBR Green I fluorescence detection (R-414 Master mix, Syntol, Moscow, Russia), and 50, 100, 200, 400, 800, 1600, 3200, or 6400 copies of genomic DNA as external standards. The calibration curve in the coordinates Ct (threshold cycle value) and minus log P (decimal logarithm of the amount of DNA standard) was plotted automatically using the LightCycler 480 System software. Gene expression is presented as the relative number of cDNA copies per 100 copies of DNA-dependent RNA polymerase 2 subunit A (*Polr2a*) cDNA, which served as an internal standard [42,43,44]. A melting-curve analysis was performed at the end of each run for each primer pair, allowing us to control the amplification specificity.

### 2.10. Western Blotting

The extraction of total protein from N1E-115 cells was performed in RIPA buffer (150 mM NaCl, 1.0% IGEPAL CA-630, 0.5% sodium deoxycholate, 0.1% SDS, and 50 mM Tris-HCl, pH 8.0) with the addition of 1 mM Na_3_VO_4_, 2 mM phenylmethylsulfonyl fluoride, and a protease and phosphatase inhibitor cocktail. The protein concentration was estimated spectrophotometrically using the Pierce BCA Protein Assay Kit (Thermo Fisher Scientific Inc., Waltham, MA, USA) and a NanoDrop 2000C spectrophotometer (Thermo Scientific, Waltham, MA, USA), followed by the adjustment of samples to equal concentrations with 2× Laemmli sample buffer. After denaturation by boiling for 10 min at 95 °C, the cell extracts (10 μg of total protein per lane) were resolved on 10% SDS-PAGE and blotted onto a nitrocellulose membrane (Bio-Rad Laboratories, Hercules, CA, USA). The membrane was incubated with a primary antibody (an anti-TrkB (1:1000; R&D Systems, cat. # AF1494, RRID:AB_2155264), anti-pTrkB (1:1000; Abcam, cat. # ab51187, RRID:AB_874043), or anti-5-HT_2A_ antibody (1:300; Abcam, cat. # ab66049, RRID:AB_1141522)) at 4 °C overnight, then the membrane was washed in Tris-buffered saline supplemented with 0.05% Tween 20 (TBS-T) and incubated with an HRP-conjugated secondary antibody. After protein detection, the blot was stripped and then re-probed with an anti-β-tubulin antibody (1:20,000; Abcam, cat. # ab6046, RRID:AB_2210370) as a loading control. Immunoreactive bands were detected by means of the Clarity Western ECL Substrate (Bio-Rad Laboratories, Hercules, CA, USA). Protein bands were documented on a C-DiGit Blot Scanner (LI-COR, Lincoln, NE, USA) and quantified in the Image Studio software (LI-COR Image Studio Software, RRID:SCR_015795, Lincoln, NE, USA). Target protein levels were assessed in chemiluminescence relative units and normalized to β-tubulin chemiluminescence relative units.

### 2.11. Statistical Analysis

The data are presented as means ± the standard error of the mean (SEM). Unless stated otherwise, the significance of pairwise differences was assessed by Student’s *t* test after a Gaussian distribution evaluation by the D’Agostino–Pearson normality test. Groupwise comparisons were made by one-way ANOVA followed by Fisher’s post hoc test. Cell groups co-expressing different amounts of 5-HT_2A_ receptors and treated with 7,8-DHF were compared by two-way ANOVA followed by Fisher’s post hoc test. In figures, significance is displayed as *p* < 0.05 (*), *p* < 0.01 (**), and/or *p* < 0.001 (***).

## 3. Results

### 3.1. 5-HT_2A_ and TrkB Receptors Form Heterodimers in Neuroblastoma N1E-115 Cells

To analyze the specific interaction between 5-HT_2A_ and TrkB, we performed co-immunoprecipitation experiments with neuroblastoma N1E-115 cells co-expressing HA-tagged 5-HT_2A_ and GFP-tagged TrkB. It is noteworthy that non-transfected N1E-115 cells express neither 5-HT_2A_ nor TrkB (data not shown). After immunoprecipitation with an antibody against the HA tag, GFP-tagged TrkB was detectable only in cells co-expressing both HA- and GFP-tagged receptors (Figure 1A). To assay the extent of artificial protein aggregation, cells expressing only one receptor type (either HA-5-HT_2A_ or GFP-TrkB) were mixed prior to lysis and analyzed in parallel. While individual receptors could be detected by the same antibody, co-immunoprecipitation did not occur (Figure 1A). This result confirmed the specificity of the 5-HT_2A_–TrkB hetero-oligomerization.

To overcome the limitations related to protein solubilization and concentration during the co-immunoprecipitation procedure, which can cause artificial protein aggregation [45], we further analyzed the interaction between 5-HT_2A_ and TrkB in living cells using a Förster resonance energy transfer (FRET)-based approach. We measured the apparent FRET efficiency (Ef_DA_) between mTurquoise2-labeled 5-HT_2A_ (5-HT_2A_-mTurquoise2, donor) and YPet-labeled TrkB (TrkB-YPet, acceptor) in living N1E-115 cells using the linear unmixing FRET (lux-FRET) method combined with confocal microscopy (Figure 1B–D). This approach can detect the physical interaction of individual molecules on the nanoscale [40]. The lux-FRET analysis revealed a high apparent FRET efficiency (Ef_DA_ = 14.6%) for 5-HT_2A_-mTurquoise2 and TrkB-YPet. In contrast, cells expressing a monomeric fluorophore-tagged CD86 protein [46] yielded significantly lower Ef_DA_ values (Figure 1D). These experiments demonstrated the selective heterodimerization of 5-HT_2A_ and TrkB.

### 3.2. Endogenous 5-HT_2A_ and TrkB Receptors Form a Protein Complex in the Mouse Brain

Having identified the interaction between recombinant 5-HT_2A_ and TrkB receptors in vitro, we next investigated whether this interaction also occurs in vivo. An immunohistochemical analysis was performed on slices of mouse hippocampus, cortex, and striatum, revealing that 5-HT_2A_ and TrkB receptors were highly co-localized (Figure 2A). We then performed a co-immunoprecipitation assay using brain tissue lysates from C57BL/6J mice and identified 5-HT_2A_–TrkB oligomeric complexes in samples from the frontal cortex, hippocampus, and striatum (Figure 2B).

As an additional highly sensitive assay of hetero-oligomerization, we performed an in situ proximity ligation assay (PLA) [47]. In all analyzed brain regions (striatum, hippocampus, and cortex), we found specific PLA-positive blobs, confirming the physical interaction between 5-HT_2A_ and TrkB (Figure 2C). The quantification revealed a significantly higher number of 5-HT_2A_–TrkB heterodimers in hippocampal cells, indicating a greater prevalence of 5-HT_2A_–TrkB heteromeric complexes in this brain region compared to in the frontal cortex and striatum (Figure 2D).

### 3.3. Calcium Signaling Is Not Affected by the Heterodimerization

To evaluate possible functional consequences of TrkB–5-HT_2A_ heterodimerization, we first assessed Ca^2+^ activity in N1E-115 cells expressing TrkB-YPet, 5-HT_2A_-mTurquoise2, or both. To this end, we used the Ca^2+^ indicator GCaMP6f [48] and analyzed Ca^2+^ dynamics with the multi-threshold event detection approach [49]. The ratio of the GCaMP6f fluorescent signal (F) to the saturated Ca^2+^ signal (F_max_) indicated that basal Ca^2+^ levels were significantly higher following heterodimerization compared to control cells (Figure 3). The addition of the Ca^2+^ ionophore ionomycin to N1E-115 cells elevated Ca^2+^ levels under all conditions, with a significantly stronger signal in cells expressing either 5-HT_2A_ alone or co-expressing TrkB and 5-HT_2A_ (Figure 3). This finding may reflect constitutive 5-HT_2A_ activity [50,51,52]. On the other hand, the basal Ca^2+^ levels did not differ between cells expressing 5-HT_2A_ and those co-expressing TrkB and 5-HT_2A_ (Figure 3), indicating that 5-HT_2A_–TrkB hetero-oligomerization did not affect the constitutive 5-HT_2A_ receptor activity toward a Ca^2+^ response. Based on this observation, we focused on whether heterodimerization had functional consequences for TrkB receptors.

### 3.4. Heterodimerization Reduces TrkB Phosphorylation and Blunts the Response to 7,8-DHF

The dimerization and autophosphorylation of tyrosine residues in the intracellular kinase domain of TrkB are crucial steps for the activation of TrkB-mediated intracellular signaling cascades [8]. To determine whether 5-HT_2A_–TrkB heterodimerization led to changes in TrkB phosphorylation, we examined the level of the phosphorylated TrkB (pTrkB) protein in N1E-115 cells, using a phospho-specific antibody that specifically recognizes the Y515 site [53]. Notably, the basal pTrkB/TrkB ratio was significantly reduced in cells co-expressing 5-HT_2A_ (Figure 4). Moreover, 5-HT_2A_ co-expression not only decreased the amount of phosphorylated TrkB but also prevented its activation by 7,8-DHF, a well-known high-affinity TrkB agonist (Figure 4A) [36,54]. We also treated cells co-expressing 5-HT_2A_ and TrkB with the selective 5-HT_2A_ receptor agonist 25CN-NBOH, which had no significant impact on the pTrkB/TrkB ratio (Figure 4B).

We then tested whether the heterodimerization rate influenced TrkB phosphorylation. To this end, we analyzed cells expressing a constant amount of TrkB (1 µg), either alone or together with an increasing concentration of HA-tagged 5-HT_2A_ (Figure 4C). Basal TrkB phosphorylation was significantly reduced by the expression of even a small amount (0.25 μg) of 5-HT_2A_ receptor. Treatment with 7,8-DHF yielded a robust enhancement of TrkB phosphorylation in cells expressing TrkB alone, and this response continuously diminished when TrkB was co-expressed with increasing amounts of 5-HT_2A_ (Figure 4B).

### 3.5. Expression Patterns of TrkB and 5-HT_2A_ during Postnatal Development

The results depicted in Figure 4 suggest that 5-HT_2A_–TrkB oligomerization impacted TrkB function in a manner depending on the receptors’ expression ratio. Therefore, we next determined the expression profiles of 5-HT_2A_ and TrkB in mouse hippocampus, striatum, and cortex at different stages of postnatal development using real-time quantitative RT-PCR. An analysis of the Ntrk2 transcript encoding TrkB revealed that the Ntrk2 mRNA level was continuously elevated from P1 to P30 in the hippocampus and frontal cortex and that this increase was sustained in the striatum until P60 (Figure 5B).

An analysis of the *Htr2a* gene transcript encoding 5-HT_2A_ revealed that the mRNA level continuously increased from P1 to P60 in the hippocampus and cortex but not in the striatum (Figure 5A). Moreover, the number of *Ntrk2* transcripts was approximately 10 times higher than that of *Htr2a* transcripts (Figure 5A). In the hippocampus and cortex, the *Htr2a*/*Ntrk2* mRNA ratio mimicked the pattern of mRNA levels, while the *Htr2a*/*Ntrk2* mRNA ratio in the striatum drastically increased from P1 to P60 (Figure 5C). These findings suggest that the balance between TrkB and 5-HT_2A_ may shift in certain brain regions during postnatal development. In light of the above-described suppressive effects of 5-HT_2A_ on TrkB phosphorylation, this balance might significantly impact TrkB functions. Considering that the physiological ratio of 5-HT_2A_ to TrkB does not exceed 0.2, we examined TrkB phosphorylation in N1E-115 cells expressing a constant amount of TrkB with near-physiological (0.01–0.25 μg) concentrations of HA-tagged 5-HT_2A_. These experiments revealed significantly reduced basal phosphorylation of TrkB in the cells expressing 5-HT_2A_ at concentrations ≥0.05 μg (Figure 5D).

### 3.6. Restoration of TrkB Phosphorylation by Ketanserin Treatment

We next analyzed whether a pharmacological 5-HT_2A_ receptor blockade could restore TrkB phosphorylation. To address this question, we pretreated N1E-115 cells expressing both receptors with the preferential 5-HT_2A_ antagonist ketanserin. It is noteworthy that after ketanserin treatment basal TrkB autophosphorylation was recovered to the level obtained in the cells expressing TrkB alone. In contrast, the 5-HT_2A_ receptor blockade by ketanserin did not restore the TrkB phosphorylation in response to 7,8-DHF treatment (Figure 6A).

To determine whether 5-HT_2A_ blockade or activation could alter TrkB autophosphorylation in vivo, we intraperitoneally injected adult (P60) C57BL/6J mice with either ketanserin (1 mg/kg) or the selective 5-HT_2A_ agonist 25CN-NBOH (1 mg/kg). TrkB phosphorylation was significantly increased in the hippocampus and frontal cortex of mice treated with ketanserin but not those treated with 25CN-NBOH (Figure 6B).

These results implied that 5-HT_2A_–TrkB heterodimerization specifically attenuates TrkB autophosphorylation and that the heterodimerization rate specifically regulates agonist-mediated TrkB phosphorylation. In vitro 5-HT_2A_ blockade with the selective antagonist ketanserin reversed the decrease of basal TrkB autophosphorylation but failed to restore the TrkB response to 7,8-DHF. At the same time, ketanserin administration considerably enhanced TrkB phosphorylation in the mouse brain in vivo.

## 4. Discussion

Several research articles have reported the modulation of BDNF expression by the 5-HT_2A_ receptor within the limbic neurocircuits [27,28,29], but the exact underlying mechanisms remain unclear. Moreover, the TrkB receptor can be involved in 5-HT_2A_-mediated neurito- and spinogenesis in cortical neuronal cultures [55]. We recently found that the chronic treatment of mice with the selective 5-HT_2A_ receptor agonists TCB-2 and 25CN-NBOH reduced the levels of total and membrane-associated TrkB protein in the mouse brain [56], indicating TrkB downregulation. Among several possible explanations for the functional interplay between 5-HT_2A_ and TrkB, heterodimerization is the most intriguing. Previously, we have reported that the 5-HT_4_ or 5-HT_7_ receptors, the classical GPCRs, can form heterodimers with adhesion molecule L1 [33], CDK5 [34], or CD44 [35]. The existence of heteroreceptor complexes between GPCRs and receptor tyrosine kinases (RTKs) was demonstrated previously for the adenosine receptor A2AR [57,58]. Moreover, the 5-HT_1A_ receptor has previously been shown to form heteroreceptor complexes with fibroblast growth factor receptor 1 (FGFR1) in the mouse hippocampus [59]. In the present study, we used a combination of multiple approaches (including lux-FRET, PLA, and co-immunoprecipitation) to demonstrate the existence of 5-HT_2A_-TrkB heteroreceptor complexes both in vitro and in vivo.

From a functional perspective, this heterodimerization suppressed basal TrkB autophosphorylation and prevented agonist-mediated TrkB activation without affecting 5-HT_2A_ receptor functions. Importantly, our data demonstrated that the degree of heterodimerization played a pivotal role in this process. This suggests that changes in the relative amounts of both receptors and the corresponding changes in heterodimerization rates can provide an intriguing mechanism for the differential regulation of TrkB functions in health and disease. Several studies have described substantially decreased TrkB expression and lower TrkB phosphorylation in the hippocampus of depressive suicidal victims [60,61,62,63]. There is also evidence of increased 5-HT_2A_ levels in some brain structures, including the frontal cortex and hippocampus, in post-mortem samples from patients with major depressive disorder and suicide victims [64,65,66]. Moreover, 5-HT_2A_ expression is reportedly sensitive to basal 5-HT concentration, with the 5-HT_2A_ receptor becoming more abundant in response to a diminished synaptic 5-HT concentration and vice versa [67,68,69]. In this regard, the general tendency of upregulated 5-HT_2A_ expression in the brain of suicide victims may be explained by an adaptive response to deficient 5-HT_2A_ signaling due to a reduced 5-HT level in depression. Based on the above-mentioned data, it might be hypothesized that under pathological conditions (e.g., major depressive disorder, addiction, and suicidal behavior), the balance between 5-HT_2A_ and TrkB becomes shifted toward increased levels of 5-HT_2A_–TrkB heteroreceptor complexes, with a subsequent loss of TrkB function. The prevalence of 5-HT_2A_–TrkB heteroreceptor complexes in the hippocampus, even under the basal conditions obtained in this study, suggests that this brain structure is among the most vulnerable to the above scenario.

Notably, our findings may also explain why the ability of BDNF to activate TrkB gradually declines during early postnatal development in mice [70]. Starting from approximately 2 weeks of age, BDNF application has only a weak influence on TrkB phosphorylation, while the systemic administration of selective serotonin reuptake inhibitors begins to affect TrkB signaling. Here, we showed that at P14 5-HT_2A_ expression was drastically increased (up to five-fold) in all analyzed brain regions, while TrkB expression exhibited only a moderate change (up to a two-fold increase). Considering that even a very low relative amount of 5-HT_2A_ receptor elicited inhibitory effects on TrkB, such a shift in the receptor ratio could result in the decreased ability of BDNF to activate TrkB.

The 5-HT_2A_–TrkB interaction could also play a role in the action of antidepressants. It is generally accepted that BDNF–TrkB signaling is a critical component of an antidepressant response [11,71,72,73]. On the other hand, accumulating evidence from animal and clinical studies suggests that 5-HT_2A_ receptor inactivation may also facilitate antidepressant action [23,74,75,76,77]. In one study, chronic treatment with a relatively high dosage (5 mg/kg) of the 5-HT_2A_ receptor antagonist ketanserin increased neurogenesis in the adult rat hippocampus [78]. However, this effect might be unspecific to the 5-HT_2A_ receptors because a combined treatment with fluoxetine and ketanserin applied at lower dosage (i.e., 0.1 mg/kg) failed to produce a neurogenic effect while it boosted the expression of *Bdnf* mRNA in the rat hippocampus [79]. Moreover, atypical antipsychotics exerting 5-HT_2A_ antagonism can stimulate BDNF expression and/or secretion [80,81,82,83,84,85,86,87,88,89,90]. On the other hand, 5-HT_2A_ receptor activation can suppress *Bdnf* transcription in the hippocampus [26]. Our present data showing the physical interaction of 5-HT_2A_ and BDNF could thus explain the functional crosstalk between these receptors at the molecular level. Indeed, we found that an acute blockade of 5-HT_2A_ with ketanserin reversed basal TrkB phosphorylation in vitro. We also demonstrated that acute treatment with ketanserin led to increased TrkB phosphorylation in the mouse hippocampus and frontal cortex in vivo, which is in accordance with our findings regarding the prevalence of 5-HT_2A_–TrkB heteroreceptor complexes in these brain regions.

Overall, our present results demonstrated that the 5-HT_2A_ receptor, a member of the GPCR family, could form heterodimers with the non-GPCR receptor TrkB, both in vitro and in vivo. Heterodimerization considerably decreased TrkB autophosphorylation and prevented TrkB activation by its ligand, and these effects could be reversed by a pharmacological blockade of 5-HT_2A_ with ketanserin. Importantly, our findings suggest that the regulated and balanced ratio of heterodimerization in different brain structures may be crucially involved in both the onset and treatment responsiveness of psychiatric diseases such as depression and anxiety.

## 5. Conclusions

In the present study, we provide a multilevel analysis demonstrating, for the first time, the physical interaction between the 5-HT_2A_ receptor, a member of the GPCR family, and the non-GPCR receptor TrkB, both in vitro and in vivo.

From a functional perspective, heterodimerization suppressed basal TrkB autophosphorylation and prevented agonist-mediated TrkB activation without affecting 5-HT_2A_ receptor functions. Importantly, these detrimental effects on TrkB functions were reversed by a pharmacological blockade of the 5-HT_2A_ receptor with ketanserin, a drug widely used to treat hypertension. Moreover, our data demonstrated that the stoichiometry of heterodimerization played a pivotal role in this process. This suggests that changes in the relative expression of both receptors and the corresponding changes in heterodimerization rates can provide an intriguing mechanism for the brain-region-specific regulation of TrkB functions in health and disease.

## Figures and Tables

**Figure 1 cells-11-02384-f001:**
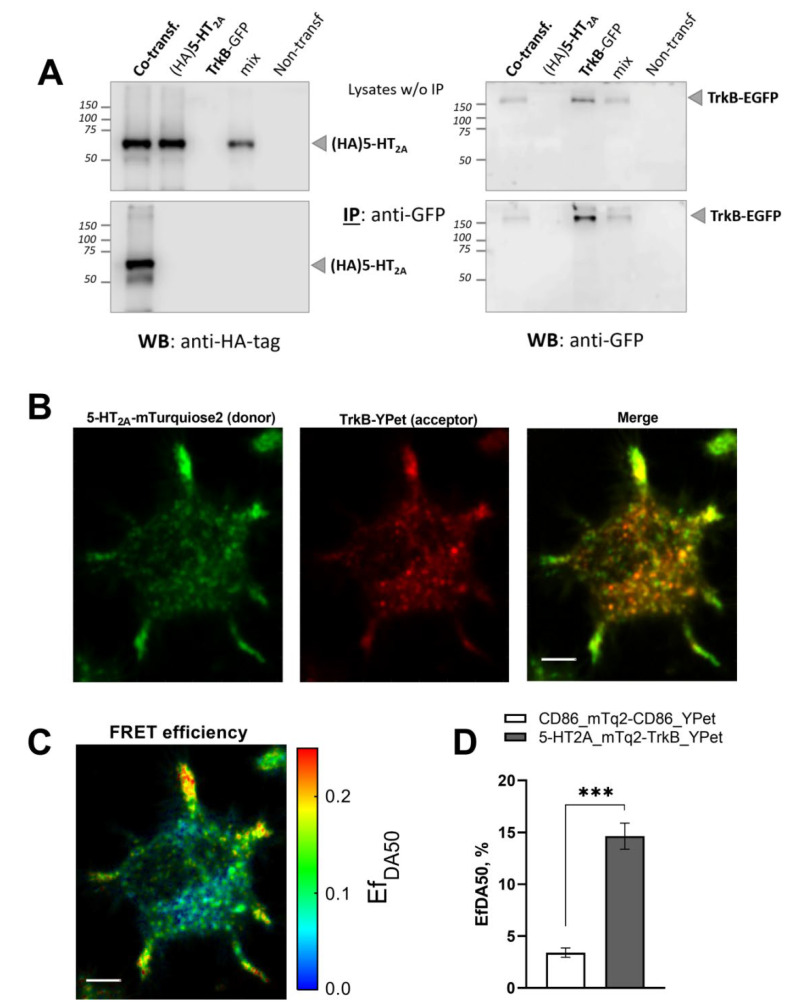
Interaction between receptors 5-HT_2A_ and TrkB in neuroblastoma N1E-115 cells. (**A**) Co-immunoprecipitation of recombinant HA-tagged 5-HT_2A_ and GFP (mEGFP)-tagged TrkB. Using a mixture of cells expressing each individual protein (mix) or cells that co-expressed both proteins (co-transfection), we performed immunoprecipitation (IP) with an anti-GFP antibody, followed by Western blot (WB) analysis with anti-GFP (right) and anti-HA (left) antibodies. Top: Expression of proteins before IP (lysate w/o IP). Bottom: Expression of proteins after IP. Results are representative of at least three independent experiments. (**B**–**D**) Specific interaction between 5-HT_2A_-mTurquoise2 and TrkB-YPet. Cells co-expressing 5-HT_2A_-mTurquoise2 and TrkB-YPet were analyzed using the lux-FRET method after confocal microscopy. (**B**) Distributions of 5-HT_2A_-mTurquoise2 (donor) and TrkB-YPet (acceptor), and merged images quantified by linear unmixing of the fluorescence emission spectra. Scale bar, 5 µm. (**C**) Apparent FRET efficiency (Ef_DA_). A representative cell is shown. (**D**) Quantification of FRET efficiency (Ef_DA_) between 5-HT_2A_-mTurquoise2 and TrkB-YPet. *** *p* < 0.001 (one-way ANOVA).

**Figure 2 cells-11-02384-f002:**
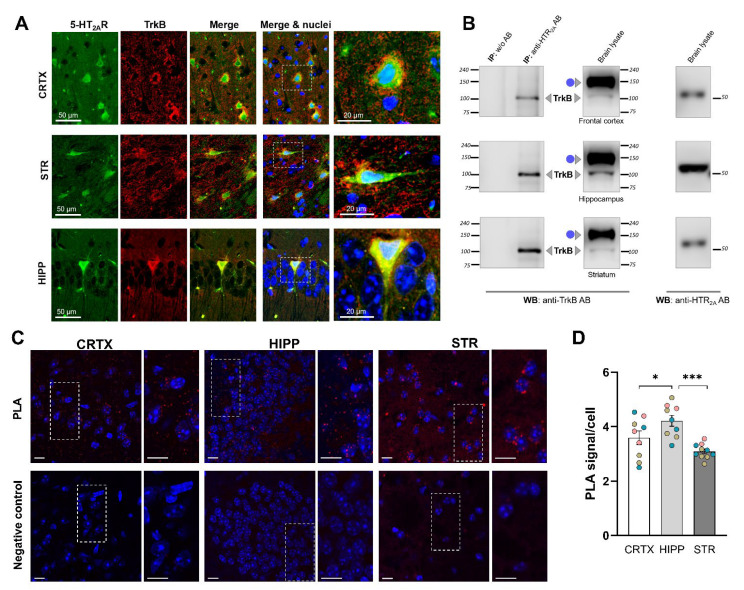
Interaction between receptors 5-HT_2A_ and TrkB in the mouse brain. (**A**) Co-localization of 5-HT_2A_ and TrkB in the mouse brain. Brain slices from cortex (CRTX), striatum (STR), and hippocampus (HIPP) were subjected to immunohistochemistry for the detection of 5-HT_2A_ (green) and TrkB (red), followed by confocal microscopy. Nuclei are shown in blue. Right: Merged images and a magnified view of representative neurons. (**B**) Co-immunoprecipitation between 5-HT_2A_ and TrkB in brain lysates. Whole-brain homogenates were prepared from different brain areas and subjected to IP with an anti-5-HT_2A_ receptor antibody, followed by WB analysis with anti-TrkB and anti-5-HT_2A_ antibodies. AB: antibody; IP, immunoprecipitation; WB, Western blot; ● glycosylated TrkB. Results are representative of at least three independent experiments. (**C**) Detection of 5-HT_2A_–TrkB heteroreceptor complexes in mouse brain slices using proximity ligation assay (PLA). Representative images of PLA staining in cortex (CRTX), hippocampus (HIPP), and striatum (STR) are shown as a single Z-stack. Nuclei are shown in blue, and PLA blobs are shown in red. Negative control was performed omitting primary antibodies. Scale bar, 20 µm. (**D**) Quantification of the 5-HT_2A_-TrkB heterodimer density. Data are presented as means ± SEM (*n* = 6 mice). Each colored dot represents data obtained for one mouse. * *p* < 0.05, *** *p* < 0.001 (one-way ANOVA).

**Figure 3 cells-11-02384-f003:**
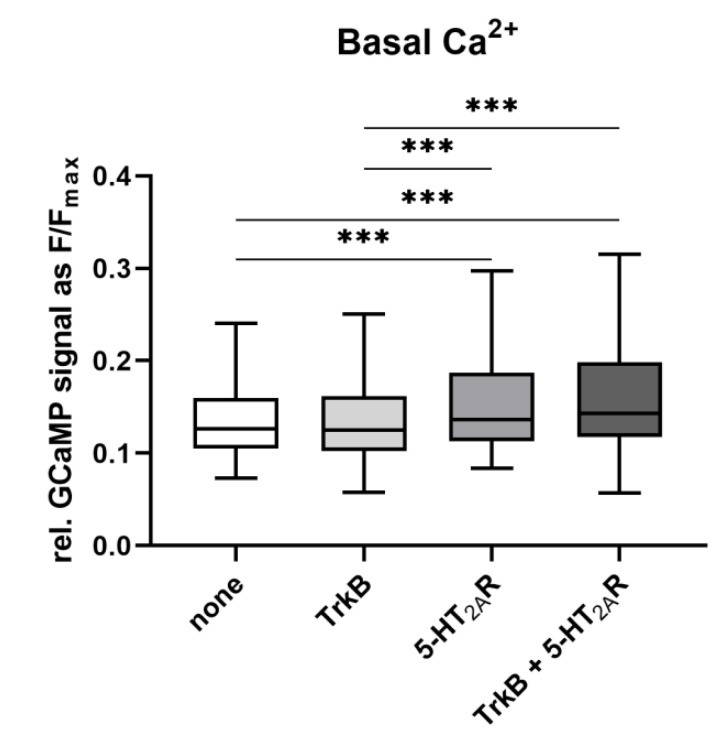
Heterodimerization does not affect basal Ca^2+^ levels in N1E-115 cells. Neuroblastoma cells were transfected with Ca^2+^ indicator GCaMP6f, either alone or together with 5-HT_2A_-mTurquoise2, TrkB-YPet, or both. Basal Ca^2+^ activity was calculated as F/F_max_. To saturate Ca^2+^ levels, 10 µM ionomycin was applied after 6 min of recording. The saturated Ca^2+^ signal was used as F_max_. Data are presented as means ± SEM (12 ≤ *n* ≤ 17). *** *p* < 0.001 (Kruskal–Wallis test with Dunn’s multiple-comparison post hoc test).

**Figure 4 cells-11-02384-f004:**
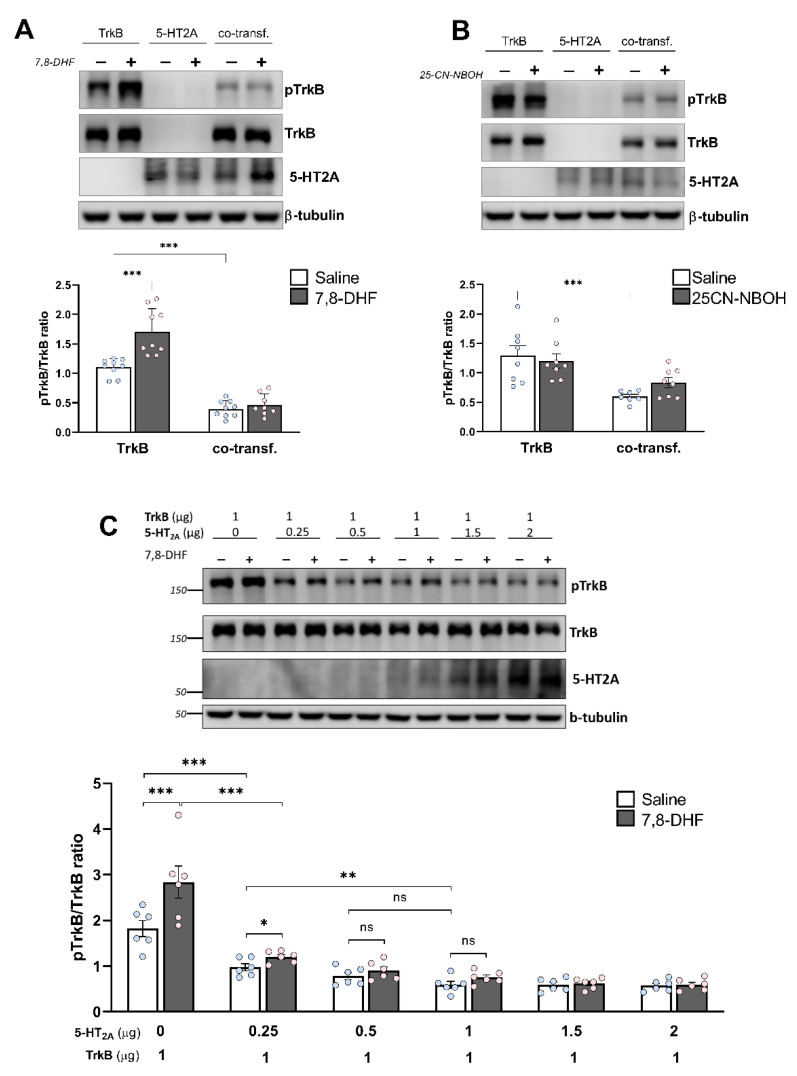
Heterodimerization affects TrkB phosphorylation. (**A**,**B**) Analysis of TrkB phosphorylation. Neuroblastoma N1E-115 cells expressing either HA-tagged 5-HT_2A_ (1 µg), GFP-tagged TrkB (1 µg), or co-expressing equal amounts of both receptors were treated for 30 min with either (**A**) the 5-HT_2A_ receptor agonist 25CN-NBOH (1 μM) or (**B**) the TrkB agonist 7,8-DHF (500 nM). Cells were lysed, followed by SDS-PAGE and Western blot (WB) analysis using antibodies against either total TrkB or phosphorylated TrkB. The 5-HT_2A_ receptor was detected in parallel. In WB, ß-tubulin was used as a loading control. Representative Western blots are shown. Lower panels show the quantification of TrkB phosphorylation, which was performed by densitometry and calculated as the ratio of total TrkB expression to the TrkB phosphorylation signal after adjustment for the general expression level. Bars show means ± SEM (*n* ≤ 8). *** *p* < 0.001 (two-way ANOVA). (**C**) Heterodimerization affects agonist-mediated TrkB phosphorylation. Neuroblastoma cells were co-transfected with 1 μg of cDNA encoding the GFP-tagged TrkB receptor, together with increasing concentrations of HA-tagged 5-HT_2A_ receptor. Cells were treated for 30 min with either 500 nM 7,8-DHF or vehicle, followed by WB analysis. Lower panel: Quantification of TrkB phosphorylation. Bars show means ± SEM (*n* = 6). ns: not significant, * *p* < 0.05, ** *p* < 0.01, *** *p* < 0.001 (two-way ANOVA).

**Figure 5 cells-11-02384-f005:**
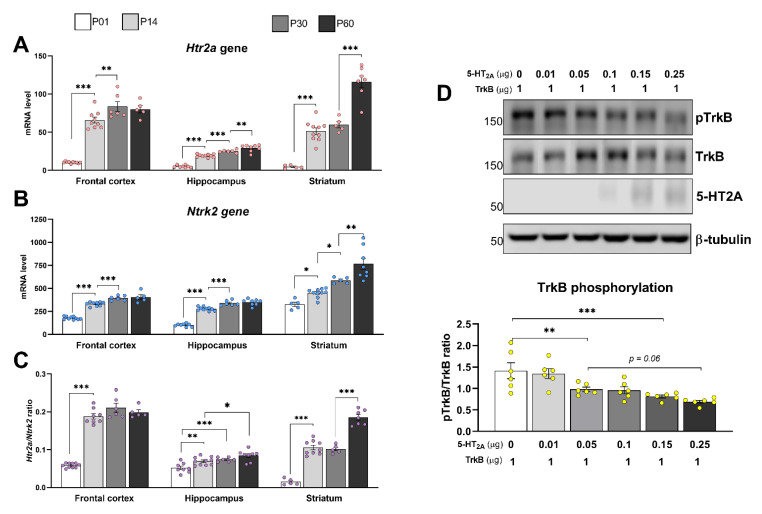
Expression profiles of TrkB and 5-HT_2A_ receptors during postnatal development. (**A**,**B**) Expression levels of mRNAs encoding 5-HT_2A_ (**A**) and TrkB (**B**) in the indicated brain regions were determined at different stages of postnatal development using real-time PCR. Gene expression is presented as the number of gene cDNA copies per 100 cDNA copies of *Polr2a*, used as a calibration control. (**C**) Expression ratios between 5-HT_2A_ and TrkB. Bars represent means ± SEM (5 ≤ *n* ≤ 10). * *p* < 0.05; ** *p* < 0.01 *** *p* < 0.001 (one-way ANOVA). (**D**) Heterodimerization exerts an inhibitory effect on agonist-independent TrkB autophosphorylation. Neuroblastoma cells were co-transfected with 1 μg of cDNA encoding the GFP-tagged TrkB receptor, together with increasing concentrations of HA-tagged 5-HT_2A_ receptor, followed by Western blot (WB) analysis. Lower panel: Quantification of TrkB phosphorylation. Bars show means ± SEM (*n* = 6). ns: not significant, *p* = 0.06: an insignificant trend; ** *p* < 0.01, *** *p* < 0.001 (one-way ANOVA).

**Figure 6 cells-11-02384-f006:**
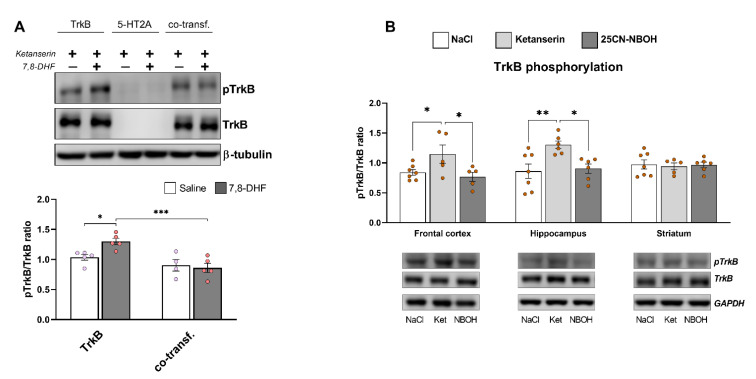
Blockade of 5-HT_2A_ with ketanserin reversed TrkB autophosphorylation in vitro and in vivo. (**A**) Neuroblastoma N1E-115 cells expressing HA-tagged 5-HT_2A_ receptor, GFP-tagged TrkB, or co-expressing both receptors were treated with the 5-HT_2A_ receptor antagonist ketanserin (1 μM) for 10 min, followed by treatment with either 7,8-DHF (500 nM) or vehicle for 30 min. Cell lysates were then subjected to SDS-PAGE and Western blot (WB) analysis with antibodies against phosphorylated or total TrkB. β-tubulin was used as a loading control. Representative WBs are shown. Bars show means ± SEM (*n* ≤ 8). * *p* < 0.05; *** *p* < 0.001 (two-way ANOVA). (**B**) Mice were intraperitoneally injected with ketanserin (1 mg/kg) or 25CN-NBOH (1 mg/kg), resulting in increased TrkB phosphorylation in the hippocampus and frontal cortex. At 10 min (for 25CN-NBOH) or 30 min (for ketanserin) after injection, the animals were euthanized, and brain lysates were prepared from the frontal cortex, hippocampus, and striatum. This was followed by SDS-PAGE and WB to detect phosphorylated and total TrkB. GAPDH was used as a loading control. Representative WBs are shown. Bars represent means ± SEM (5 ≤ *n* ≤ 8) * *p* < 0.05; ** *p* < 0.01 (two-way ANOVA).

**Table 1 cells-11-02384-t001:** The primer sequences, annealing temperatures, and PCR products’ lengths.

Target Gene	Primer Sequences	Annealing Temperature, °C	Amplicon Length, bp
*Ntrk2*	F 5′-cattcactgtgagaggcaacc-3′ R 5′-atcagggtgtagtctccgttatt-3′	63	175
*Htr2a*	F 5′-agaagccaccttgtgtgtga-3′ R 5′-ttgctcattgctgatggact-3′	61	169
*Polr2a*	F 5′-gttgtcgggcagcagaatgtag-3′ R 5′-tcaatgagaccttctcgtcctcc-3′	63	188

## Data Availability

Data supporting the findings of this study are available from the corresponding authors on reasonable request.

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
