# Peer review of "Serotonin Receptor 5-HT2A Regulates TrkB Receptor Function in Heteroreceptor Complexes"

_cells, 2022, doi:10.3390/cells11152384_

Round 1

Reviewer 1 Report

Manuscript Cells-1796035-v1: "Serotonin receptor 5-HT2A Regulates TrkB Receptor Function in Heteroreceptor Complexes" by Tatiana Ilchibaeva et al.

In this manuscript, the authors hypothesized that the serotonin 5-HT2A receptor and tropomyosin receptor kinase B (TrkB) interactions may regulate development and pathophysiological mechanisms. The authors report interactions between 5-HT2A and TrkB receptors in neuroblastoma N1E-115 cells and in the mouse brain using coimmunoprecipitation, FRET, and proximity ligation assay approaches. The authors further show that heterodimerization does not affect basal Ca2+ levels in N1E-115 cells, but decreases TrkB autophosphorylation, preventing its activation with agonist 7,8-DHF, even in the presence of low 5-HT2A receptor expression. They also show postnatal developmental regulation of 5-HT2A and TrkB receptor expression in the cortex, hippocampus, and striatum. Using the non-selective antagonist ketanserin to block 5-HT2A receptors, the authors found no downregulation of TrkB phosphorylation, and no restoration of the TrkB response to its agonist 7,8-DHF in vitro. In adult mice, they report that intraperitoneal injection of the 5-HT2A receptor antagonist ketanserin (1 mg/kg) (with no effect of 25CN-NBOH 1 mg/kg) increases basal TrkB phosphorylation in the frontal cortex and hippocampus, coherent with the prevalence of 5-HT2A–TrkB heteroreceptor complexes in these brain regions. The authors conclude that functional role of 5-HT2A–TrkB receptor heterodimerization, via regulated expression of 5-HT2A and TrkB receptor is a molecular mechanism for brain region-specific modulating TrkB functions during development and under pathophysiological conditions. 

The results presented in this manuscript are sound and interesting. However, there are several issues that need to be clarified before it can be published.

1-The use of "selective" for the 5-HT2A receptor agonist 25CN-NBOH or antagonist ketanserin is not correct. Initial characterization of these compounds showed that 25CN-NBOH has barely 20-fold higher affinity for 5-HT2A than for 5-HTB or 5-HT2C receptors (Halberstadt Neuropharmacology 2016, 107, 364–375), and similarly ketanserin has about 10-fold higher affinity for 5-HT2A than for 5-HT2C receptors (Boess & Martin Neuropharmacology 1994 vol. 33 pp. 275-317). Furthermore, ketanserin has also fairly high affinity for alpha1 adrenergic and histamine receptors. They should be called "preferential" antagonist or agonist. A dose of 0.1 mg·kg-1·day-1 ketanserin was used in a previous study to avoid possible effects on 5-HT2C receptors (ref 76, Pilar-Cuéllar et al., British Journal of Pharmacology 2012, 1651046–105). It would be safer if the authors used lower concentration of these drugs or better to use more selective drugs e.g. MDL100907 as selective 5-HT2A receptor antagonist.

2-The 5-HT2A receptor is not really a classical 5-HT receptor identified as implicated in neurogenesis or depression as suggested in the introduction. The authors are quoting reference 75 (Jha et al., Neurosci. Lett. 2008, 441, 210–214) that report anti proliferative effect of ketanserin using 5 mg/kg, which, at this dose, can block several other  receptors. The other quoted reference (ref 76, Pilar-Cuéllar et al., British Journal of Pharmacology 2012, 1651046–105) using 0.1 mg·kg-1·day-1 of ketanserin did not found neurogenic effects, supporting a non-specific dose effect of ketanserin. Furthermore, the authors are presenting "5-HT2A-mediated neurogenic effects" by quoting reference 55 (Ly et al., Cell Rep. 2018, 23, 3170–3182), which presents in fact neuritogenesis and/or spinogenesis effects. The authors should revise both introduction and discussion of these papers.

3-Are all experiments done on cell lines performed in serum-free condition to avoid serotonin eefct from the serum that could interfere with the experiments?

4-Why are the antibodies against 5-HT2A receptors used for immunoprecipitation different from the one used for immunofluorescence or western blots, (Santa Cruz Biotechnology, cat. # sc-15073, RRID:AB_2119724) vs. (Abcam, cat. # ab66049, RRID:AB_1141522)? Have these antibodies been tested for specificity on 5-HT2A KO mice for example?

-Did the authors tested for endogenous expression of receptors 5-HT2A and TrkB receptors in neuroblastoma N1E-115 cells? This information should be added and eventually discussed in respect to the results.

-A quantification of RT-qPCR data should be presented.

-A dot representation should be used for graphs with low n numbers.

Author Response

Reviewer#1

1) The use of "selective" for the 5-HT2A receptor agonist 25CN-NBOH or antagonist ketanserin is not correct. Initial characterization of these compounds showed that 25CN-NBOH has barely 20-fold higher affinity for 5-HT2A than for 5-HTB or 5-HT2C receptors (Halberstadt Neuropharmacology 2016, 107, 364–375), and similarly ketanserin has about 10-fold higher affinity for 5-HT2A than for 5-HT2C receptors (Boess & Martin Neuropharmacology 1994 vol. 33 pp. 275-317). Furthermore, ketanserin has also fairly high affinity for alpha1 adrenergic and histamine receptors. They should be called "preferential" antagonist or agonist. A dose of 0.1 mg·kg-1·day-1 ketanserin was used in a previous study to avoid possible effects on 5-HT2C receptors (ref 76, Pilar-Cuéllar et al., British Journal of Pharmacology 2012, 1651046–105). It would be safer if the authors used lower concentration of these drugs or better to use more selective drugs e.g. MDL100907 as selective 5-HT2A receptor antagonist.

Reply:

We fully agree with Reviewer that ketanserin is not the highly selective 5-HT2A receptor (5-HT2AR) antagonist. Therefore, we replaced term "selective" by “preferential” throughout the text of manuscript. In contrast, 25CN-NBOH seems to be specific 5-HT2AR agonist. Indeed, data presented in the recent review of Røsted et al., (Chem Med Chem 2021, doi: 10.1002/cmdc.202100395) demonstrate that 25CN-NBOH display 37-fold 5-HT2AR/5-HT2BR selectivity and 52- to 100-fold 5-HT2AR/5-HT2CR selectivity. For example, widely used 5-HT2AR agonist DOI has only 3.4-7.5-fold selectivity in comparison to 5-HT2CR (May et al., J. Pharmacol. Exp. Ther., 2003, vol.306, pp.301-309; Nelson et al., Naunyn Schmiedebergs Arch Pharmacol., 1999, vol. 359, pp.1-6). Moreover, the selective 5-HT2C antagonist SB-242084 has been shown to do not affect head twitch response (HTR) evoked by 1.5-mg/kg dose of 25CN-NBOH (Buchborn et al., Front Pharmacol, 2018, doi: 10.3389/fphar.2018.00017). This data suggests that at low to medium doses 25CN-NBOH preferentially binds to 5-HT2A receptors in vivo. As we used 1.0 mg/kg of 25CN-NBOH in our in vivo experiment, we could exclude the confounding effects of 5-HT2C receptors.

In the N1E-115 cells transiently expressing recombinant 5-HT2A receptor, the majority of problems associated with selectivity of drugs were avoided because these cells do not express any serotonin G-protein coupled receptors endogenously. 

2) The 5-HT2A receptor is not really a classical 5-HT receptor identified as implicated in neurogenesis or depression as suggested in the introduction. The authors are quoting reference 75 (Jha et al., Neurosci. Lett. 2008, 441, 210–214) that report anti proliferative effect of ketanserin using 5 mg/kg, which, at this dose, can block several other receptors. The other quoted reference (ref 76, Pilar-Cuéllar et al., British Journal of Pharmacology 2012, 1651046–105) using 0.1 mg·kg-1·day-1 of ketanserin did not found neurogenic effects, supporting a non-specific dose effect of ketanserin. Furthermore, the authors are presenting "5-HT2A-mediated neurogenic effects" by quoting reference 55 (Ly et al., Cell Rep. 2018, 23, 3170–3182), which presents in fact neuritogenesis and/or spinogenesis effects. The authors should revise both introduction and discussion of these papers.

Reply:

As recommended by the Reviewer, we have revised quoting references and discussed the findings presented in these works more accurate (p.12, line 404-405; p.13, lines 454-459).

3) Are all experiments done on cell lines performed in serum-free condition to avoid serotonin effect from the serum that could interfere with the experiments?

Reply:

We apologize for not being sufficiently clear here. In all our experiments, N1E-115 cells were incubated in a serum-free medium after transfection. We mentioned that in “Materials and Methods” section of revised manuscript (p. 2, lanes 89-90).

4) Why are the antibodies against 5-HT2A receptors used for immunoprecipitation different from the one used for immunofluorescence or western blots, (Santa Cruz Biotechnology, cat. # sc-15073, RRID:AB_2119724) vs. (Abcam, cat. # ab66049, RRID:AB_1141522)? Have these antibodies been tested for specificity on 5-HT2A KO mice for example?

Reply:

This has a simple technical explanation. We used commercially available anti-5-HT2A antibody #sc-15073 from Santa Cruz for co-immunoprecipitation (co-IP) since only this antibody was validated by the manufacturer for co-IP application. However, since this antibody were produced in goat, it was not possible to apply it in the immunofluorescence analysis for double staining with TrkB because anti-TrkB antibodies were also sourced from goat. For this reason, we have chosen Abcam anti-5-HT2A antibodies #ab66049 sourced from rabbit. The latter antibody was also used in the Western blot analysis. According to the manufacturer’s data, both antibodies were tested on knockout animals for their specificity. Moreover, our experiments in non-transfected N1E-115 cells, which endogenously express neither TrkB nor 5-HT2A receptors revealed no specific signal after staining with these antibodies (see also our response to point 5 below).     

5) Did the authors tested for endogenous expression of receptors 5-HT2A and TrkB receptors in neuroblastoma N1E-115 cells? This information should be added and eventually discussed in respect to the results

Reply

We thank the Reviewer for this important note. We tested native N1E-115 cells for endogenous expression of TrkB and 5-HT2A receptors both by quantitative RT-PCR (see figures R1 and R2 below) as well as by Western blot (see Figure R3 below). In both cases, we did not obtained any endogenous expression of these proteins.  We mentioned this finding in the revised manuscript at page 5, lines 224-225.

Figure R1. Amplification curves (A) and melting peaks (B) after RT-PCR assessment of TrkB transcripts in samples from native, non-transfected N1E-115 cells and samples from the frontal cortex of adult mice.

Figure R2. Amplification curves (A) and melting peaks (B) after PCR assessment of 5-HT2A transcripts in samples from native N1E-115 cells and samples from frontal cortex of adult mice.

Figure R3. Representative images of WB analysis with anti-TrkB antibodies (A) or anti-5-HT2A antibodies from Abcam (#ab66049) (B) in N1E-115 cells expressing either HA-tagged 5-HT2A or GFP-tagged TrkB receptors, or co-expressing equal amounts of both receptors.

6) A quantification of RT-qPCR data should be presented.

Reply:

There must have been some misunderstanding there. In Figure 5 A to C of initial manuscript we already presented quantification of qRT-PCR experiments. The PCR data are shown as the relative number of cDNA copies per 100 copies of DNA-dependent RNA polymerase 2 subunit A (Polr2a) cDNA. In revised manuscript, we added detailed description of RT-PCR data quantification to the “Materials and Methods” section (p.4, lines 180-186).

4) A dot representation should be used for graphs with low n numbers.

Reply:

As recommended by the Reviewer, we added dots to graphs where the n number was low

Reviewer 2 Report

In the present manuscript, Ilchibaeva et al. present evidence that the serotonin receptor 5-HT2A can regulate TrkB receptor function in heteroreceptor complexes. Specifically, the authors newly demonstrate physical interactions between these two different types of receptors.

In all, the manuscript is written with expertise and the research group involved has already published several papers dealing with the functions of 5-HT2A receptors. The findings of the study are indeed novel and potentially relevant. The methods applied sound mostly smooth, and the literature discussed is relevant. However there are some issues mainly concerning the use of 5-HT2A receptor antibodies and speculations in the discussion that should be carefully addressed by the authors.

Major points: The authors use two different polyclonal antibodies against 5-HT2A in their experiments, one from Santa-Cruz for the Co-immunoprecipitation experiments and one from Abcam for the immunofluorescence assays.

1.) In general, immunohistochemistry using antibodies against G protein-coupled receptors is quite tricky and possesses several shortfalls like low specificity. Thus, the authors should provide much more details about the specificity of the 5-HT2A receptor antibodies used in their study. This information might come from control-experiments (Western blot analysis, pre-incubation with the antigen), but also from the literature and/or the respective antibody datasheets. It would also be informative to provide a low-power photomicrograph showing 5-HT2A labeling in the frontal cortex and/or hippocampus to compare the immunofluorescence staining produced by their rabbit anti-5-HT2A with data published in the literature (e.g. Weber & Andrade (2010) https://doi.org/10.3389/fnins.2010.00036

2.) Given the broad but region-specific distribution of 5-HT2A receptors and TrKB throughout the brain, some parts of the discussion, specifically those about a possible functional role of 5-HT2A –TrkB interactions in the actions of antidepressants and atypical antipsychotics (between lines 452 and 467) should be toned-down, since many other non-molecular but circuit-specific factors are involved in the actions of these drugs.

Minor points: Abstract, first sentence, should read: ….and tropomyosin receptor kinase B (TrkB) strongly contributes to neuroplasticity regulation and are implicated in      

Line 346: The authors state that they performed RT-PCR using tissue from mouse hippocampus, striatum and cortex, which parts of cortex where used? Frontal cortex as shown in Figure 5, or also other parts of cortex. The authors should specify this.

In the discussion, the authors might provide other examples of physical interactions between GPCR and non-GPCR-type receptors.

Author Response

Reviewer #2

Major points: The authors use two different polyclonal antibodies against 5-HT2A in their experiments, one from Santa-Cruz for the Co-immunoprecipitation experiments and one from Abcam for the immunofluorescence assays.

1) In general, immunohistochemistry using antibodies against G protein-coupled receptors is quite tricky and possesses several shortfalls like low specificity. Thus, the authors should provide much more details about the specificity of the 5-HT2A receptor antibodies used in their study. This information might come from control-experiments (Western blot analysis, pre-incubation with the antigen), but also from the literature and/or the respective antibody datasheets. It would also be informative to provide a low-power photomicrograph showing 5-HT2A labeling in the frontal cortex and/or hippocampus to compare the immunofluorescence staining produced by their rabbit anti-5-HT2A with data published in the literature (e.g. Weber & Andrade (2010) https://doi.org/10.3389/fnins.2010.00036

Reply:

We thank the Reviewer for this important note. As mentioned in our response to the Reviewer#1, we have used commercially available anti-5-HT2A antibody #sc-15073 from Santa Cruz for co-immunoprecipitation (co-IP) since only this antibody was validated by the manufacturer for co-IP application. However, since this antibody were produced in goat, it was not possible to apply it in the immunofluorescence analysis for the double staining with TrkB (anti-TrkB antibody was also sourced from goat). For this reason, we have chosen Abcam anti-5-HT2A antibodies (#ab66049) sourced from rabbit. The latter antibody was also used in the Western blot analysis. According to the manufacturer’s data, both antibodies were tested on knockout animals for their specificity. Moreover, our experiments in non-transfected N1E-115 cells, which endogenously express neither TrkB nor 5-HT2A receptors revealed no specific signal after staining with this antibody (Fig. R1 below).  

We also followed Reviewer’s advice and compared low-resolution immunofluorescence images produced in our study using rabbit anti-5-HT2A from Abcam (Fig. R2 below) with data published by Weber and Andrade (Front. Neurosci, 2010), who also used Abcam antibody in their study (Fig. 4D in above-mentioned publication). Here we obtained a similar distribution pattern with the 5-HT2AR mostly expressed on cell somata and partly on the neuropil.

Figure R1. Representative images of WB analysis with anti-TrkB antibodies (A) or anti-5-HT2A antibodies from Abcam (#ab66049) (B) in N1E-115 cells expressing either HA-tagged 5-HT2AR or GFP-tagged TrkB receptors, or co-expressing equal amounts of both receptors.

Figure R2. Brain slices from the frontal cortex of adult (P90) C57BL/6J male mouse were subjected to immunohistochemistry for the detection of 5-HT2A (green) using Abcam anti-5-HT2A antibodies (#ab66049) sourced from rabbit followed by confocal microscopy. Nuclei are shown in blue.

2) Given the broad but region-specific distribution of 5-HT2A receptors and TrKB throughout the brain, some parts of the discussion, specifically those about a possible functional role of 5-HT2A –TrkB interactions in the actions of antidepressants and atypical antipsychotics (between lines 452 and 467) should be toned-down, since many other non-molecular but circuit-specific factors are involved in the actions of these drugs.

Reply:

As recommended by the Reviewer, we relativized our statement concerning the possible role of 5-HT2AR-TrkB heterodimerization. In the revised manuscript (page 12, lanes 454-463), we highlighted that heterodimerization could explain a functional 5-HT2AR-BDNF crosstalk, that in turn could be involved into both antidepressants and antipsychotic drugs action.

Minor points: Abstract, first sentence, should read: ….and tropomyosin receptor kinase B (TrkB) strongly contributes to neuroplasticity regulation and are implicated in 

Reply:

We corrected this sentence accordingly    

Line 346: The authors state that they performed RT-PCR using tissue from mouse hippocampus, striatum and cortex, which parts of cortex where used? Frontal cortex as shown in Figure 5, or also other parts of cortex. The authors should specify this.

Reply:

In the revised manuscript, we specified that RT-PCR was performed in samples isolated from the frontal cortex.

In the discussion, the authors might provide other examples of physical interactions between GPCR and non-GPCR-type receptors.

Reply:

As suggested by the Reviewer, we have provided additional examples of physical interaction between GPCRs and non-GPCR receptors in “Discussion” section (p.12, lines 410-415).
